# The Impact of Online Theological Studies during the COVID-19 Period on Students' Religiosity/Spirituality: A Qualitative Analysis

Viktória Šoltésová *◯ and Marek Harastej

Faculty of Education, Matej Bel University, 974 11 Banská Bystrica, Slovakia; marek.harastej@student.umb.sk
* Correspondence: viktoria.soltesova@umb.sk

**Abstract:** Our study wants to clarify the structure of spirituality by applying existing multidimensional theories of religiosity and spirituality to in-depth interviews conducted among a sample of students. The current research analyzes 15 qualitative semi-structured interviews conducted among a sample of students at the Adventist Theological Institute in the Czech Republic and was conducted in May 2022. Since religiosity is a multidimensional phenomenon and we wished to investigate development in each dimension, we based our analyses on Glock and Stark's model with four of their dimensions of religiosity: "belief", "practice", "experience", and "knowledge". Our study reflects on existing multidimensional religiosity/spirituality; six dimensions by Huber overlap with the religiosity/spirituality model we chose as the applied multidimensional model. As a result, based on the interviewees' statements, we distinguished these dimensions in the context of specific conditions in the respondents' individual personal experiences in the absence of personal group interaction. In our research, we ask the question: "What impact has the move of the entire formal and informal environment of an educational institution to an online environment had on the spirituality of each student?" An important finding is that the COVID-19 pandemic period brought about an exciting stimulus for spiritual support in theological education. The emergence of individual and independent religiosity/spirituality is a significant religious change.

**Keywords:** religiosity; spirituality; COVID-19 pandemic period; theological education; online environment



## 1. Introduction

The research aimed to map the issue of the development of religiosity/spirituality in specific conditions without direct social interaction and in the conditions of higher theological studies. Examining religiosity and spirituality simultaneously has become an important area of research on religious individuals and communities. Research on religiosity and spirituality has progressed recently, also thanks to investigation of them as a social phenomenon during the COVID-19 pandemic period. The work of the German sociologist (Glock and Stark 1965a), followed up (Halman and De Moor 1993) in the work *Religion, Churches, and Moral Values*, tries to describe the dimensions of religiosity. A structure of spirituality formed by applying multidimensional theories of religiosity and spirituality to in-depth interviews and the original application of qualitative interview data are used in (Demmrich and Huber 2019). The extended six-dimensional model by Huber can be applied to almost all of the spirituality-relevant codes. Therefore, in principle, the scope of this six-dimensional model can be expanded to spirituality. Their findings, discussed within the framework of Huber's Centrality of Religiosity, extend the concept of religiosity to spirituality without mutually excluding these concepts.

*1.1. Approaches to Religiosity/Spirituality*

This qualitative study applied a semi-structured interview guideline of religiosity/spirituality to a stratified sample of N = 15 students (age 18–21) of theology in the Czech Republic. Adventist Theological Institute (2022). The impact of an educational institution in an online environment throughout the COVID-19 pandemic on the spirituality of each student gave us interesting explanations of the importance of social interaction in theological education. The main goal was to analyze the dimensions of religiosity among religious students, confirm their occurrence in the observed sample of Christian students, and answer the question of the subjectively perceived need for social interaction in theological education. Approaches and directions in religiosity/spirituality research are indicated in the following parts. The orientation towards the absolute is common to both *religiosity* and *spirituality*. However, religiosity is considered a more collective and dogmatic phenomenon. Spirituality, on the other hand, tends to be described as a more individual and experiential matter related to sensitivity to spiritual values in general (Hay and Nye 1998). Spirituality has been a much-studied phenomenon of religion in psychology and sociology over the past two decades, but it lacks conceptual, structural, and operational coherence (Demmrich and Huber 2019). Wong (2012) refined (Frankl 1992)'s notion of the noetic dimension of personality (Frankl used the terms "spiritual" and "noetic" as synonyms, and he did not understand the term "noetic" in a religious sense) and define the three basic dimensions of human existence: physical, psychological, and spiritual, while "the noetic dimension of personality lies in the area of the intermingling of the psychological and the spiritual dimensions". According to this concept, the spiritual dimension includes the consciousness of the spiritual sphere, contact with the transcendent, and the ability to know God. The noetic dimension includes Frankl's "will to meaning," moral judgment, and spiritual beliefs and values. Using a multidimensional approach, religious involvement cannot be defined simply in terms of various frequency indicators, such as frequency of church attendance or prayer. Sociologist Glock (Adamson et al. 2000) presented a model of dimensions of *religiosity*: beliefs (belief-inner religiosity), practiced religiosity (action, prayer, and fasting), knowledge and intellect (cognitive aspect), experiential religiosity (based on the experiencing aspect), and the consequences aspect of religiosity (effects of religiosity). Glock and Stark (1965b)'s model is not very recent, but it provides a useful structure for the highly complex phenomenon of religiosity, dividing it into different categories referred to as dimensions of religiosity. The authors' five dimensions of religiosity were later reduced to four by dropping the 'consequences' dimension. These dimensions can further be broken down into sub-dimensions. Allport relates the distinction between intrinsic and extrinsic religiosity to personality predispositions Stríženec (1996). Typologically, religiosity is divided into intrinsic and extrinsic (Allport 1950; Ragan and Malony 1976). These models are well-suited for illustrating the complexity of religiosity/spirituality. For the most part, *spirituality* is nowadays divided into religious, natural, and humanistic. Humanistic spirituality is considered a universal phenomenon. MacDonald (2000), based on factor analysis of data from several scales of spirituality, found the following five dimensions of spirituality: 1. cognitive orientation to spirituality, 2. experiential–phenomenological dimension, 3. existential well-being (meaning of life), 4. paranormal beliefs, and 5. religiosity (focus on intrinsic religiosity as opposed to extrinsic religiosity). Some researchers refer to spirituality as a personality dimension (Emmons 1999). Emmons introduced the concept of spiritual intelligence, which expresses people's sensitivity to transcendent realities. Rather, as *The Guardian* states, spirituality is "an individual, personal, direct, and experiential awareness of the transcendent dimension, a certain value orientation with self-respect, other people, nature, and the absolute/God" (Stríženec 2001). Říčan (2005) presented five factors of spirituality in Czech society based on the research of his team. Apart from the diachronic aspect, there are different approaches to the definition of spirituality in psychology, philosophy, and theology. Smékal (2004), in connection with the psychology of personality, states that by the spiritual dimension of its being, the personality transcends its everyday life, while at the same time opening itself to the formation of such characteristics as love and

responsibility. According to (Smékal 2004), spirituality is a path to an ethically grounded life, an instrument of personality change, a way of restoring peace in the soul, and a way of salvation. Spirituality should be appropriate to the personality characteristics of each person. Slovak psychologist (Halama 2007) relates spirituality to the search for the meaning of life. Also, from our *theological* perspective, every day lived spirituality is more significant than exploring the subjective semantics of spirituality. Our study therefore follows a different approach, as it wants to clarify the structure of spirituality by reflecting on existing multidimensional theories of religiosity (Huber and Huber 2012) and spirituality through in-depth interviews conducted among a sample of students. We followed (Glock and Stark 1965b)'s dimensions of religiosity:

"belief", "practice", "experience", and "knowledge ".

*1.2. Research on Religiosity/Spirituality*

Research on religiosity proceeds in the same way as research on other psychological phenomena, but it does not clarify all aspects. However, there is also a current tendency to understand religiosity as a multidimensional phenomenon manifesting itself in non-religious behavior as well. In our context of Central and Eastern Europe (CEE), various scales related to research on religion and religiosity have begun to be used by Stríženec et al. They also tested the following methodologies: the Searching Scale, the Religious Fundamentalism Scale (Altemeyer and Hunsberger 1992), the Religious Coping Styles (Pargament 2007), and the Religious Conflict Scale (Funk 1958). Slovak's translation of the shorter Expressions of Spiritual Inventory (ESI) scale from 2000, authored by MacDonald, was validated by (Stríženec 2007). Also, the validity of the Spiritual Transcendence Scale (STS), authored by R. L. Piedmont, was verified in our conditions (Stríženec 2007). Říčan (2005) compiled a spirituality questionnaire, which they called the Prague Spirituality Inventory, as part of their research at the Psychological Institute of the Academy of Education of the Czech Republic. The relationship between the psychic and spiritual aspects of man is explained in several ways. Psycho–spiritual dualism separates spirituality from the mental aspect of personality. Spiritual reductionism considers all problems that do not have a physiological basis to be spiritual. Psychological reductionism, on the other hand, regards all problems as either psychological or somatic. It reduces spirituality to psychology so that spiritual experience is nothing more than a kind of psychological mechanism. In real research, however, the manifestations of religiosity cannot be strictly separated. It is difficult to draw a line between what is psychological and what stems from a spiritual source.

The investigation of religiosity based on personality–psychological criteria was started by the authors: (Fukuyama 1961; Glock and Stark 1965a; Boss-Nünning 1972). Their work resulted in theories and typologies of religiosity. In their research, Tloczynski et al. (1997) found that the level of spirituality correlates with characteristics of a healthy personality. The following methods were used on a research sample of 145 university students: Spiritual Orientation Inventory (SOI), Dimensions of Religious Ideology Scale (DRIS), Personal Orientation Inventory (POI), and Minnesota Multiphasic Personality Inventory-2 (MMPI-2). Interviews as a research method in the study of respondents' religiosity was also used by (Russell and Spilka 1967). Concerning religiosity, Chlewinski (1981) defined the most important positive functions of religion in personality as follows:

- The meaning of life—religion's response to existential questions and satisfaction of the need for transcendence;

- Value system—providing basic moral values and criteria for decision-making, the direction of a person's efforts in conformity with certain values;

- Security and trust-fulfilling—the need for one's own identity and psychological integrity;

- Self-identification and identification with the group—satisfying the need for interpersonal relationships;

- Self psychotherapy—providing meaning in life and removing sources of neurotic tensions, freedom from guilt.

We are interested in research that reveals the power of internalized religiosity to the point of disinterest in religious content. Based on the outcome of religious socialization research in Hungary (Pusztai and Demeter-Karászi 2019), those conveniently keeping and building autonomous religiosity, the religiously impaired, those drawing near, and the unreflective nostalgic types belong to the religious field of their reconstruction model of religious socialization.

## 2. Results

### 2.1. Semi-Structured Interview Basic Analysis

2.1.1. Question 1 (with Sub-Questions Answered Separately)—How Do You Perceive Your Spirituality: 1. Your Relationship with God, 2. Your Prayer Life, 3. Your Time for Studying the Bible, and 4. Your Time for Contemplation, in the Last Two Years?

These were open-ended questions with no limit on the length of answers. Although in the respondents' answers, satisfaction and a positive outlook prevailed, when freely evaluating their spirituality in words, about half of the comments explain or excuse the negatives (lack of time, difficult family or work situation, health problems, the word "busy" repeated twice, etc.). The other half, implicitly assuming improvement, explain it by a deeper insight into the biblical text, by experiences lived or shared, or as a positive consequence of the study. As noted above, the most common word used spontaneously by respondents is the word "time". It occurs six times in total and only once in a positive sense (improvement due to study, I read the Bible more, I try to make time for personal relationships with God every day) and five times in a negative sense (the time I have to divide between work, school, family; deterioration due to less time; family and life situations, time; less time for meditation due to more responsibilities). This can be supplemented by five other statements that, although they do not explicitly mention time, refer to it significantly as a problematic point (I am very overwhelmed with tasks…; working in a place where I have to do everything "right away"; too many responsibilities; heavy workload; busy). Based on the above, it can be concluded that the problem of time is a significant majority problem of the respondent population, i.e., students in the area of their spirituality. The paradox is that despite the positive or rather positive assessment of their spirituality, only three respondents gave an explicitly positive answer to the questions about the cause of their current state. Positive responses are as follows:

- Many new spiritual stimuli. Lots of new books read and more frequent prayers (also because of the COVID-19 pandemic, the war in Ukraine, deaths of acquaintances).
- Various crises and seeking God's will, which brings me closer to God.
- Experiences from teachers, sermons, and books.
- Through studying more often reading the Bible and thinking more deeply about spiritual issues.

2.1.2. Question 2—What Role Has the Adventist Theological Institute Played in Your Spiritual Life?

This was again an open question with no limit on the length of the answers. When asked about the direct influence of the Adventist Theological Institute (ATI), all but three of the negative or rather negative comments (I don't read what I want to, but what I have to; it has taken away some of my time for God, but has changed my perspective on some things—for example, different perceptions of other churches—I was previously strictly opposed; it has taken away my time and ability to focus on God) are significantly outweighed by the positive evaluations. Spontaneously, six respondents reported that ATI was beneficial to their spirituality in the form of deeper knowledge, expanded knowledge, or greater understanding of the biblical text (four respondents mentioned the Bible explicitly several

times). All of these perspectives are united, we might say, by the cognitive aspect of spiritual life enrichment. Four respondents spontaneously mentioned people and community in various forms, and thus the enrichment of spirituality through relationships. And also, four respondents mentioned that ATI helped them to be directed to service. Some answers:

- It gives me an understanding of new depths and dimensions, mainly due to the knowledge and analysis of the texts that we do. Then when I read the Bible, it speaks to me more. Still, personal time with God is key for me.
- Deeper insight into the biblical text, shared discussions, and a diverse community of students and teachers.
- Clarification of many theological problems.
- It positively broadens my horizons and brings me closer to the Lord God.

### 2.1.3. Question 3—How Does ATI help You Most in Spirituality?

The eighth question duplicates question seven to some extent. Again, in the form of an open-ended question, it returns to the role of ATI in respondents' spirituality, this time with a clear focus on the area in which they perceive a contribution. Two of the 15 respondents do not perceive a contribution of ATI to their spirituality, 1 cannot identify, and the remaining 12 respondents name the following areas in which they believe ATI helps them in spirituality (some respondents—ATI helps: to understand Bible better; another finding contexts; communicating with people; understanding different kinds of spirituality; engaging in ministry; helping to think about relationships with others and the right to one's opinion; practical live, sharing, community, common direction; examples of other people's spirituality; in approaching the spirituality of biblical characters, etc.). Some answers:

- In a better understanding of the Bible.
- In communication. In building relationships.
- The self-study subject of spirituality is also nurturing.
- The spirituality of biblical characters and notables as inspiration.

### 2.1.4. Question 4—In What Way Does ATI Not Help (Harm) You Most in Spirituality?

In another open-ended question, we asked for the opposite opinion, i.e., in what ways respondents perceived the negative impact of ATI on their spirituality. Six respondents returned to the already mentioned issue of time (it takes time—but this is not a problem of spirituality (direct); it takes time after studying concentration to other things with God is low; studying takes me much time...; time-consuming, especially the obligatory reading; sometimes it happens that because of the amount of studying, there is no time to develop a relationship with God). Four responded "I don't know", and three gave other reasons (missing the opportunity to talk to a spiritual in a more personal way, otherwise I can't say anything bad; too much emphasis on information, and disappointment with some teachers' opinions; negative examples of some teachers' personal lives; classmates losing their faith at ATI). We can therefore conclude that no recurring problem has emerged, except for the rather indirect ATI-related problem of lack of or poor time management.

### 2.1.5. Question 5—What Impact Did the COVID-19 Period Have on Your Spirituality?

While the first question asked for the respondents' perception of their spirituality over the last two years without any mention of COVID-19, this question explicitly mentions this phenomenon inherent in the last two years. Students related the last two years to a period including mainly the last more or less problem-free months in this area, whereas those asking about the COVID-19 period associated the period more with a period defined by strict lockdown, etc. It can be concluded that, with some minor variation, the two questions are more or less the same in their results.

2.1.6. Question 6—What Kind of Stimuli did ATI Give You for Your Spirituality during the COVID-19 and in Which Forms?

This was an open-ended question. Of the fifteen respondents, one did not respond and two others responded that ATI did not give them any incentives during COVID-19. Two others more or less minimized the influence (minimal because we were not together and online was rather killing me; not much because online is not enough for understanding), one considered the influence of ATI to be the same as outside COVID-19, two expressed gratitude for at least being online, and others acknowledged more or less the influence of online learning on their spirituality (spiritual sharing with classmates and teachers had a very positive influence on my spirituality; I didn't know exactly what to expect, just all the stimuli was beneficial; new books; online discussions and teaching helped me think about some things—interesting guests showed me different ways of thinking about old topics; term papers + tutorials—new perspectives; especially the study of Old Testament books where we delved deeply, it gave me new perspectives on God, etc.).

2.1.7. Question 7—What Stimuli and Forms for Your Spirituality during the COVID-19 Did You Miss Most from ATI?

The following question surveyed what they perceived to miss from ATI's influence in fulfilling their spirituality. This was again an open-ended question. The most common response was personal contact and fellowship, which appeared 10 times among the responses in various forms. This was followed by a request for something more practical, mission, Bible reading together, or a more practical focus on the post-COVID period.

2.1.8. Question 8—Your Suggestions for Spiritual Formation at ATI during Any Further Lockdown and Periods of Online Teaching

This was followed by a final question, an open-ended question, allowing respondents to give ATI input for the next period of potential restrictions. This question was left unanswered by most respondents to previous questions. There were a few declarations against COVID-19 (hope this will not be repeated; take every opportunity to be together and encourage each other), followed by a few specific suggestions.

*2.2. Summary of Research Results*

The first part of the entire survey was designed to focus on respondents' perceptions of their spirituality over the past two years, with no emphasis on COVID-19. We consequently paired them with the dimensions of religiosity according to the (Glock and Stark 1965a) model: beliefs (belief-inner religiosity), practiced religiosity (action, prayer, and fasting), knowledge and intellect (cognitive aspect), and experiential religiosity (based on the experiencing aspect). A code of one of the four dimensions is applied; if not applicable, a new dimension should be formed; the numbers of the given codes are counted and compared with each other (frequency analysis). All K = 513 given codes of implicit or explicit spirituality and religiosity were completely absorbed by the four dimensions of (Glock and Stark 1965a)'s model (Table 1). Except for face-to-face communication, sub-dimensions were formed within every dimension: "belief", "practice", "experience", and "knowledge". Issues related to the promotion of the development of religiosity by the staff of the educational institution without personal contact with students during the lockdown period of the COVID-19 pandemic were reflected separately.

**Table 1.** Frequency analysis of dimensions and sub-dimensions of spirituality according to (Glock and Stark 1965a) model (K = 513).

| Dimensions and Sub-Dimensions | k | % |
|---|---|---|
| **Beliefs** | **104** | **19.9** |
| higher spiritual being/God | 74 | 14 |
| meaning of life | 6 | 1.2 |
| discussions | 14 | 2.7 |
| human being/teleology | 1 | 0.2 |
| others | 9 | 1.8 |
| **Practiced/Ritual religiosity** | **76.6** | **30** |
| *Public practice* | *58* | *11.4* |
| church service and sermons | 13 | 2.5 |
| community and relationships | 21 | 4.1 |
| practical ministry | 17 | 3.4 |
| holiday celebrations | 1 | 0.2 |
| ecumenism | 2 | 0.4 |
| *Private practice* | *95* | *18.6* |
| relationship with God | 43 | 8.4 |
| prayer life | 31 | 6 |
| time for contemplation | 12 | 2.4 |
| others | 4 | 0,8 |
| **Knowledge and intellect (cognitive aspect)** | **173** | **33.8** |
| Intellect | 44 | 8.6 |
| discussions with others | 10 | 2 |
| studying Bible | 17 | 3.3 |
| studying theological literature | 14 | 2.3 |
| formal education | 67 | 13 |
| questioning the meaning of life | 5 | 1 |
| theodicy | 7 | 1.4 |
| reflexivity | 9 | 1.8 |
| **Experiential religiosity** | **67** | **16.3** |
| Lived experiences | 36 | 7 |
| Shared experiences | 22 | 4.3 |
| New spiritual experience | 1 | 0.2 |
| Ethics in relationships | 5 | 1 |
| others | 3 | 0.6 |
| **Others (out of the dimensions)** | **16** | **3.2** |
| **Total** | **513** | **100** |

The dimensions and sub-dimensions as the answers in the process of semi-structured interviews are mentioned continually in the part with the research results. When we summarize it, we can see several specifics regarding the selected sample of respondents in individual dimensions. The dimension of Beliefs (covering 19.9% of the total codes) includes belief in a higher spiritual being/God, searching for the meaning of life, discussions about their faith, and teleology. The sub-dimension "others" reflects the codes that were reported less than 1.8% of the time within the Beliefs dimension, which were faith in the meaning of things, the order of creation, etc. Likewise, the dimension of Practiced/Ritual religiosity (covering 30% of the total codes) reflects the importance of the dimension in the respondents' statements. It is divided into Public practice and Private practice. Sub-dimension Practiced religiosity/Private practice/Relationship with God has a frequency of 8.4%. The dimension of Knowledge and intellect (33.8%) overlaps the cognitive aspect and represents central cognitive challenges: for example, reflexivity (reasoning about one's spiritual beliefs for their plausibility and logical consistency). The Experiential Dimension (16.3%) concerns "all those feelings, perceptions, and sensations which are experienced by an actor or defined by a religious group as involving some communication, however slight, with a divine essence" (Glock and Stark 1965a). The Others (3.2%), not included in the dimensions, have an overlap in the field of spirituality and are related to statements related to the COVID-19 pandemic. We were interested in the religiosity dimensions of the respondents, specifically in the context of the period of study when they lacked mutual personal interaction during the online only study of theology at the time of the COVID-19 pandemic. In the first question, respondents expressed a predominantly positive view of their spirituality over the period that overlaps with their studies at ATI. In the following sub-questions, which

made the first question specific to particular areas of spirituality, respondents expressed predominantly positive views of their prayer life. A slightly more skeptical view was then expressed about time for Bible reading and time for contemplation and meditation. In a question that asked about the reasons for this despite previous positive assessments, about half of the respondents took a somewhat defensive apologetic position and emphasized the negatives. In a follow-up question on the role of ATI in this perception, positive comments predominated. Some respondents perceived benefits in the cognitive aspect of spiritual enrichment and a few in the enrichment of spirituality through relationships and orientation to service. In questions that asked more specifically about the benefits or, conversely, the problems associated with ATI's contribution to their spirituality, 12 of the 15 respondents cited specific benefits to their spirituality. Among the negatives, the problem of time, caused in part by studying at ATI, was sovereignly predominant. Question 6 and the following questions dealt explicitly with COVID-19 and related online learning and its impact on students' spirituality. Question 6, focusing on spirituality in general during COVID-19, more or less overlapped with Question 1 with a slight negative bias. Asking about the impact of studying at ATI during COVID-19 on their spirituality, despite responses still being more on the positive end of the spectrum, we saw results shift significantly in the negative direction. It is worth noting here that this is still the same period of two years when all respondents were ATI students, which they rated as highly beneficial to their spirituality in the previous questions, and they rated their spirituality significantly positively in the previous two years. This was followed by open-ended questions inquiring about respondents' spiritual stimuli during COVID-19 and missing ones in the online learning course. The limitation of our study is that the 15 interviews do not provide enough information to get a sense of the direction of changes in the religiosity/spirituality of a generation of theological students in the Czech Republic. Consequently, there is a need for further interviews to more accurately test the criteria put forth by our analysis, and quantitative investigations also need to be carried out. An important finding is that the COVID-19 period brought about an exciting stimulus for spiritual support in theological education. The emergence of the individual and independent religiosity/spirituality is already a significant religious change.

## 3. Materials and Methods

### 3.1. COVID-19 Period and the Associated Transition to Distance Education

The COVID-19 period and the associated transition to distance education was a profound intervention in the educational process at all levels of education. We are currently reflecting on this period with some distance. Theological education has its specifics, which have an impact on the educational process itself. The theological formation is one of these specifics, i.e., a targeted pedagogical formative influence on the religious experiences of students. This study examines the influence of distance education specifically on the theological formation of students through a case study of one particular educational institute. The Adventist Theological Institute was chosen for the research. The Adventist Theological Institute had, in years 2020–2021, 22 students in the combined theology program. This study program is delivered in the form of self-study combined with three-day study consultations held once a month. There is also a weekly intensive course once a year, and this is supplemented by two or three hours of online instruction once a week. So in practice, under standard conditions, they spend approximately 37–40 days of face-to-face contact with other students each year. In addition to the 30 days spent in consultations and 7 days in the intensive course, students can add 1–3 days when they meet at various forms of church professional conferences, etc. During these days, they not only have classes, but they also have morning and evening devotions, worship, prayer times, and various unorganized discussions and conversations on spiritual topics during breaks, walks together, or free time. As a result of the COVID-19 pandemic, they were forced to move their classes online from March 2020 to March 2022. The summer intensive courses and two consultations were exceptions. The mitigation of restrictions allowed students and staff to hold these

activities with an in-person presence. Some of the students were from the Czech Republic and some were from the Slovak Republic. Different approaches to contact and travel restrictions did not allow them to hold face-to-face meetings, although it was possible in some of the countries. Generally speaking, the students spent exactly two years with various forms of meeting restrictions, and the standard 37–40 days of face-to-face meetings were reduced to 10–12 days. The consultations continued, including teaching and common devotions and services, but they moved to the online space, specifically, in their case, to the ZOOM platform. This research explores the impact of this move to the online space on the spirituality of the students. In our research, we ask the question: "What impact has the move of the entire formal and informal environment of an educational institution to an online environment had on the spirituality of each student?" In the models of religiosity, we looked at the saturation of each category and its sub-categories in the respondents' statements. Thus, we were interested in the level of religiosity/spirituality in these theology students and their self-assessment of the changes that were associated with moving all activities that develop these aspects in the community to non-contact teaching.

### 3.2. Research Method

For the research, we used an interview as a qualitative research method used to collect primary data. It involved asking respondents about their opinions on the topic of religiosity/spirituality. This method allowed researchers to obtain detailed information that might not be available through other research methods. The semi-structured interview contained eight open-ended questions. The document was completed during the consultation in May 2022. All 15 students present participated as respondents. The respondents were from two classes: second year and fourth year. The fourth-year students had been completing the four-year study program; therefore two years of study were completed in the regular mode and two years were completed in the COVID-19 period, which was more or less, with exceptions, online learning. The students of the second year experienced only an intensive course and one tutorial in-person, and then they were forced to study the whole time in online mode except for a summer break. Thus, the two-year period in research coincides with the period when all students studied at the Adventist Theological Institute. The set of questions used for semi-structured interviews was (cited literally in the Section 2):

- The first set of questions (Q1 with sub-questions) were on religiosity during the COVID-19 period, and the questions were worded as follows:

How do you perceive your spirituality (your relationship with God), prayer life, and your time for studying the Bible (and time for contemplation) in the last two years?

- The second set of questions (Q2-Q4) was formulated in the context of support during the study of theology at the selected institution where the research was conducted and contained these questions:

What role has the Adventist Theological Institute played in your spiritual life?

- The third set of questions (Q5-Q8) was on the influence of COVID-19 on the development of spirituality in personal life as well as in connection with the study of theology:

What impact did the COVID-19 period have on your spirituality?

What kind of stimuli did ATI give you for your spirituality during the COVID-19 and in which forms?

As researchers, we created these interview questions to answer our basic research questions:

- Our first research question was whether the phenomenon of religiosity/spirituality development during the COVID-19 period can be a new form of religious participation, which makes the individuals formulate their relation to religiosity more individually and independently.

- Our second research question was whether reconstructing the religiosity/spirituality development process surfaces in the course of the young generation's religious socialization during the study of theology.

We analyzed (Glock and Stark 1965b) four dimensions

- belief, practice, experience, and knowledge,

individually to determine the dimensions that weakened in the period without direct socialization and those that were strengthened.

## 4. Discussion

The current study was conducted to examine the multidimensionality of religiosity and spirituality by comparing the applicability of (Glock and Stark 1965a)'s model of religiosity to comprehensive qualitative interview data. This qualitative study applied a semi-structured interview guideline of religiosity/spirituality to a stratified sample of N = 15 students of Theology in the Czech Republic. In the study, Demmrich and Huber (2019)'s explicit spirituality and religiosity were completely absorbed by the six dimensions of (Huber 2003), which overlap with the model we chose as the applied multidimensional model of religiosity/spirituality in our research (Glock and Stark 1965a). Dimension Public Practice received up to 12%, which is similar to our model (11.4%). Dimension Private Practice is a dimension much more saturated in our sample of respondents. The explanation is that those researched were religious persons who have decided to study theology, and part of the curriculum in their studies is explicitly the development of their relationship with the transcendent. Huber's Intellect dimension has the sub-dimension of "visiting sacred places" in the study by (Demmrich and Huber 2019), which does not occur among students from a non-Catholic environment. The Other sub-dimensions as the part of beliefs are similar to those in our research, and this dimension is saturated by 19.38%. It is absent in Glock and Stark's dimension Beliefs and our research dimension Knowledge and beliefs (cognitive aspect) covering 33.8% of the total codes, also. Glock and Stark's dimension of Experiential religiosity in our research (16.3%) overlaps Huber's dimension of spiritual "experience", which covers about eight of the total codes (Demmrich and Huber 2019) not divided into sub-dimensions. Our sub-dimensions are lived experiences, shared experiences, new spiritual experiences as a consequence of the study, ethics in relationships, and others. In our research, we raised the question of what effect the shift of teaching and also the entire formal and informal environment of an educational institution to an online environment had on the religiosity/spirituality of individual students. Most important for answering this question is the mutual relations between Question 1 (How do you perceive your spirituality (your relationship with God) in the last two years?) and Question 2 (What role has ATI played in your spiritual life?) as well as question Question 6 (What kind of stimuli ATI give you for your spirituality during the COVID-19 and which forms?). Even though the questions asked about the same period (the period of the two previous years overlapping more or less with the COVID-19 period, and this overlapping with the period of the respondents' online teaching at ATI) and the same discipline (spirituality), the individual responses are different. The emergence of the individual and independent religiosity/spirituality is an important research result of this study. Why do questions cover essentially the same period and discipline? It appear to be two possible explanations. The first is the cognitive bias due to the negative view of COVID-19 and its associated lockdown and online communications. For a part of society, including our respondents, this has become a highly emotive issue without the ability for rational detachment. Thus, as long as they answered the questions asked after the previous two years without a specific connection to COVID-19, positive evaluations prevailed. The moment COVID-19 was explicitly mentioned, the evaluation shifted in a negative direction. This reflects the general mood in society, which, of course, is necessarily reflected in the views of our students. However, this view is contradicted by a significant shift in the mean values, which reflects the overall mood of the entire sample of respondents. Cognitive bias due to negative perceptions of the COVID-19 period is not excluded, but this explanation does not seem

fully satisfactory. A second explanation would be the answer that is echoed in the responses to Question 7. We are thinking here about the lack of forms and incentives in the era of online learning. Here, the emphasis on face-to-face encounters and personal interaction with teachers and other students emerged for ten of the fifteen respondents. This may be a genuine majority perception where, despite the technological possibilities for most people, interpersonal contact is essential for the formation of spirituality and its healthy development. The question remains, of course, whether this is a generational specificity or a general setting of human beings. Several dependent and independent variables come into play, and we worked with data from individuals who answered questions about themselves. Only further research and reflection on the period we have gone through during the COVID-19 pandemic will answer our questions.

## 5. Conclusions

The individual exploration of the dimensions of religiosity proved valuable during the research, as each dimension is characterized by distinct intensities and dynamics. In our research, we asked the question: "What impact has the move of the entire formal and informal environment of an educational institution to an online environment had on the spirituality of each student?" The answer to our research question based on the research conducted is as follows. Moving teaching and the entire formal and informal environment of an educational institution to an online environment does not have any destructive effect on the spirituality of these students. The majority of interviewed students rated their spirituality significantly positively just as they positively rated the educational institution's contribution to their spirituality over these two years. However, when asked directly about the impact of online learning on their spirituality, their positive outlook shifted more toward the average, and in open-ended questions, they repeatedly reported that they lacked interpersonal contact to develop their spirituality. Based on the data above, this view may be due to cognitive bias or groupthink syndrome, but a more likely explanation seems to be the real need most people have for interpersonal contact to shape their spirituality. Individual exploration of the dimensions of religiosity has proved useful. In conclusion, the results support that (Glock and Stark 1965a)'s model of religiosity can be used on comprehensive qualitative interview data to examine religiosity and spirituality as well, which supports the approach of spirituality including religiosity. Distancing from the events of the COVID-19 period prevented students from social interaction with students and teachers of theological studies. At the same time, we are interested in the opinions of spiritualists and pastoral workers in all theological institutions in the Czech Republic to get a view from the other side and from providers of theological education who declare the development of the religiosity/spirituality of their students as the goal of their curriculum. A comprehensive analysis of these data is going to take place next year.

**Author Contributions:** Conceptualization, V.Š.; methodology, V.Š.; software, V.Š.; validation, V.Š. and M.H.; formal analysis, V.Š.; investigation, M.H.; resources, V.Š.; data curation, V.Š.; writing—original draft preparation, M.H. and V.Š.; writing—review and editing, V.Š.; visualization, V.Š.; supervision, V.Š.; project administration, V.Š.; funding acquisition, V.Š. All authors have read and agreed to the published version of the manuscript.

**Funding:** This research received no external funding.

**Institutional Review Board Statement:** Ethical review and approval were waived for this study because the research presents no more than minimal risk of harm to subjects and involves no procedures for which written consent is normally required outside the research context. The ethical aspect is part of the Ph.D. admission procedure when the applicant submits a research project for approval. The Code of Ethics of Matej Bel University in Banská Bystrica following Paragraph 8 section 14 letter c of the Statute of the MBU, in the meaning of Paragraph 9 section 1 letter a point 1. of Law No.131/2002 (Higher Education Law) of the Collection of Laws on Higher Education and on Amendments and Additions to Certain Acts as amended, contains Article 5 Ethical Principles in Scientific and Research Activities. In point 6, staff and students undertake: "In the research in

which they participate, they shall guarantee voluntary participation, respect human dignity and the research itself shall be conducted in such a way as to avoid psychological or physical harm to the participants". This UMB Code of Ethics has been approved by the Academic Senate of the UMB following Paragraph 9 section 1 letter a point 1. of the Higher Education Law. The research for the article was approved based on the submitted dissertation research project, which was carried out as part of the doctoral studies at Matej Bel University, where the author Dr. Habil. PaedDr. Viktória Šoltésová, PhD. works as a supervisor and co-author M.A. Marej Harastej is a Ph.D. student. But we have collected the data also under the supervision of our Official School Counselling Center that provided the Informed Consent. That Consent covers the info about GDPR. In paragraph 26 it is: "The principles of data protection should therefore not apply to anonymous information, namely information which does not relate to an identified or identifiable natural person or to personal data rendered anonymous in such a manner that the data subject is not or no longer identifiable. This Regulation does not therefore concern the processing of such anonymous information, including for statistical or research purposes". So, it seems that this allows me to use these data also for research purposes as it is all anonymous.

**Informed Consent Statement:** Not applicable.

**Data Availability Statement:** The datasets generated during and/or analyzed during the current study are available from the corresponding author upon reasonable request.

**Conflicts of Interest:** The authors declare no conflicts of interest.

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
