# Peer review of "The Impact of Online Theological Studies during the COVID-19 Period on Students’ Religiosity/Spirituality: A Qualitative Analysis"

_religions, doi:10.3390/rel15040500_

Round 1

Reviewer 1 Report

Comments and Suggestions for Authors

I think the work should be published (with some editing) due to its relevance in the field of spirituality research. My concerns have to do with the readability of the text. The article mentions further research and that's a positive move forward. As I read the article, I wonder about the impacts on the spirituality of these students due to the war in Ukraine and the deaths of loved ones. The article states that, in one way, the on-line environment was not a hindrance to the spirituality of these students. In another way, the absence of interpersonal engagement was difficult for them. It would be good to analyze what is personal and what is social about the spirituality of human beings. 

Comments on the Quality of English Language

Readability refers to the readers' freedom to follow the semantic flow of the text.

1. Please clarify: you mention 3 themes, 4 dimensions, 7 questions and yet there are 8 questions that you asked. The relationship between these aspects of the research would show up more clearly if you put the text into paragraphs and used bullet points to separate items you want readers to focus upon. For example, using bullet points in lines 204-215 allows the reader to focus on these points and reflect on them. I mention these lines but it would help the reader to use paragraphs and bullet points from the beginning of the text.

Paragraphing and using bullet points allows the reader to digest smaller pieces of the text so as to understand it more fully and see the relationship between the ideas presented.

There are some incomplete sentences in the text. These show up best if the sentences are read aloud. See for example, lines 32-37.

line 59: identify Frankl

line 123: use gender inclusive language

line 190: omit the second used

lines 146-147: I don't understand the expression "to the point of disinterest"

line 214: punctuation

line 194: references 7 questions

line 203 references 3 themes

line 334 references 8 questions

Again, I think clarity would be improved by paragraphs and bullet points to distinguish these aspects of the article.

Author Response

Thank you for your comments. We made several changes and added to the body of the article to address these comments.

  1. I divided the text (lines 204-2025) with the research method into paragraphs and clarified the distribution of the groups of questions, which are literally quoted in the following third chapter with the results of the research.
  2. I clarified the difference between the questions set for the interviews and what we call the research questions that we asked ourselves as researchers and tried to find out the answers to.
  3. lines 32-37 The sentence has been rephrased.
  4. line 59: We added a citation: Frankl, V. E. Man's search for meaning: An introduction to logotherapy. Beacon Press: Boston, MA, 1992.
  5. line 123, line 190: The sentence has been rewritten.
  6. line 146-147: The sentence has been supplemented by the explanation.
  7. line 214: The questions have been split, and we have adjusted their layout and punctuation as a result.
  8. line 194: Corrected, there were eight open-ended questions.
  9. line 203: Deleted.
  10. line 334: there were eight open-ended questions.
  11. The text has passed an additional language correction.

We would like to thank the following for guidance in the distribution of the description of the research results. The study is clearer and the results are more precisely defined.

Reviewer 2 Report

Comments and Suggestions for Authors

The article The Impact of Online Theological Studies in COVID-19 Period on Students’ Religiosity/Spirituality: Qualitative Analysis contains an abstract with the keywords, introduction, materials and methods, results, discussion, and conclusion. The paper aims to ask the question "What impact has the move of the entire formal and informal environment of an educational institution to an online environment on the spirituality of each student?"

The introduction briefly places the study in a broad context. The author states that examining religiosity and spirituality has become an important area of research on religious individuals and communities. As an argument refers to relevant research from this area. In the two separate parts, the author describes approaches to religiosity/spirituality and then describes relevant existing research on religiosity/spirituality. Although the author problematized differentiation between religiosity and spirituality, she/he stays by the approach of Glock and Stark. In the materials and methods, the author states the specific theological study and briefly introduces the reader to the situation at The Adventist Theological Institute in the years 2020-2021. The author used an interview as a qualitative research method and collected students' answers about opinions on a topic of religiosity/spirituality. The interviews were articulated around three themes. The first questions were on religiosity during the Covid-19 period, the second set of questions was formulated in the context of support during the study of theology at the Adventist Theological Institute where the research was conducted, and the third set of questions was on the influence of Covid-19 on the development of spirituality in personal life as well as in connection with the study of theology. The author presented a basic analysis of semi-structured interviews. In some parts, there are cited answers from students who participated in the interview. The article contains one table that represents frequency analysis of dimensions and sub-dimensions of spirituality according to Glock and Stark's (1965) model. The dimensions are beliefs, practiced/ritual religiosity, knowledge and intellect (cognitive aspect), experiential religiosity, and others (out of the dimensions), and all of them are commented on. The table properly shows the data, and it is easy to interpret and understand it.  The data are interpreted appropriately and consistently throughout the manuscript. 

Based on qualitative analysis of the sample of 15 students of the Adventist Theological Institute in the Chech Republic, the author concluded that moving teaching from an educational institution to an online environment does not have any destructive effect on the spirituality of students. 

The article is clear, relevant to the field, and presented in a well-structured manner.

The cited references are mostly recent publications, that are relevant to the topic, and the author refers to the articles published in Religions.

The weakness of the article is the small number of respondents for conclusion on the national level but the author is aware of that fact. The author opens an opportunity for further research on the same method, but a different population of spiritualists and pastoral workers in all theological institutions in the Czech Republic.

The article The Impact of Online Theological Studies in COVID-19 Period on Students’ Religiosity/Spirituality: Qualitative Analysis is interesting for the readership of the Journal, especially for the readership of special issues Child and Adolescent Spirituality/Religiosity and Religious Education which is focused on the description of concepts and the validation of new or already established questionnaires to measure specific aspects of spirituality/religiosity.

The content of the article matches the title very well. The reference list is organized alphabetically (please, pay attention to line 522 – the author Chlewinski – maybe move to line 510, and please check where in the article is quoted  Russell – specified in line 533). Cited publications are relevant to the topic of the article.

Author Response

Thanks you for your support and kind words. I have made a few changes based on the comments.

I have integrated your notes in the revised version:

  1. line 522: Chlewinski has been moved in the reference list according to the rules of alphabetical order.
  2. Current line (in adopted document) 132: Russell and Spilka have been added in the text as authors who used interview as a research method in the study of respondents' religiosity.
